# Cryo-EM structures of type IV pili complexed with nanobodies reveal immune escape mechanisms

David Fernandez-Martinez[1,7], Youxin Kong[1,6,7], Sylvie Goussard[1], Agustin Zavala [1], Pauline Gastineau [1], Martial Rey [2], Gabriel Ayme [3], Julia Chamot-Rooke [2], Pierre Lafaye [3], Matthijn Vos[4], Ariel Mechaly[5] & Guillaume Duménil [1] ✉

Type IV pili (T4P) are prevalent, polymeric surface structures in pathogenic bacteria, making them ideal targets for effective vaccines. However, bacteria have evolved efficient strategies to evade type IV pili-directed antibody responses. *Neisseria meningitidis* are prototypical type IV pili-expressing Gram-negative bacteria responsible for life threatening sepsis and meningitis. This species has evolved several genetic strategies to modify the surface of its type IV pili, changing pilin subunit amino acid sequence, nature of glycosylation and phosphoforms, but how these modifications affect antibody binding at the structural level is still unknown. Here, to explore this question, we determine cryo-electron microscopy (cryo-EM) structures of pili of different sequence types with sufficiently high resolution to visualize posttranslational modifications. We then generate nanobodies directed against type IV pili which alter pilus function in vitro and in vivo. Cryo-EM in combination with molecular dynamics simulation of the nanobody-pilus complexes reveals how the different types of pili surface modifications alter nanobody binding. Our findings shed light on the impressive complementarity between the different strategies used by bacteria to avoid antibody binding. Importantly, we also show that structural information can be used to make informed modifications in nanobodies as countermeasures to these immune evasion mechanisms.

Throughout evolution, living organisms have selected specific means of interacting with their environment. For instance, prokaryotes have macromolecular nanomachines on their surfaces that perform multiple functions important for their survival. Type IV pili are filamentous nanomachines that exemplify such bacterial surface structures[1,2]. Typically, the type IV pili are helical polymers constituted of one protein component, the major pilin. The assembly and disassembly of the major pilin leads to rapid pilus extension and retraction within a few seconds. Such dynamic behavior is governed by a bacterial nanomachine composed of about 15 proteins located across the inner and outer bacterial membrane[3]. Pathogenic *Neisseria* species, including *Neisseria gonorrhoeae* and *N. meningitidis* are classical examples of type IV pili harboring bacteria. Type IV pili not only serve as multifunctional organelles allowing the adherence to host cells and the

[1]Institut Pasteur, Université Paris Cité, INSERM UMR1225, Pathogenesis of Vascular Infections, 75015 Paris, France. [2]Institut Pasteur, Université Paris-Cité, CNRS, UAR 2024, Mass Spectrometry for Biology, 75015 Paris, France. [3]Institut Pasteur, Université Paris-Cité, CNRS-UMR 3528, Antibody Engineering Platform, 75015 Paris, France. [4]NanoImaging Core Facility, Center for Technological Resources and Research, Institut Pasteur, 75015 Paris, France. [5]Institut Pasteur, Crystallography Platform-C2RT, CNRS-UMR 3528, Université Paris Cité, Paris, France. [6]Present address: Sanofi R&D, Integrated Drug Discovery, CRVA, 94403 Vitry-sur-Seine, France. [7]These authors contributed equally: David Fernandez-Martinez, Youxin Kong. ✉e-mail: guillaume.dumenil@pasteur.fr

auto-aggregation of these bacteria[2], but also play an essential role in their pathogeneses[4,5]. In this paper, *Neisseria meningitidis* is used as a model system.

Given their crucial functions in pathogenesis and abundance as surface structures, type IV pili represent highly promising targets for vaccines, diagnostic tools, and therapeutic antibodies[6,7]. In the specific context of pathogenic *Neisseria spp.* attempts to use them as vaccine targets have, however, been abandoned early on with the realization that the surface of these structures is highly diverse between strain and can vary within a given strain[7]. *Neisseria* species have developed an arsenal of strategies to evade immune responses targeting type IV pili[8] (Supplementary Fig. 1a). These strategies can be placed in three broad groups as depicted below.

First, regarding pilin genes themselves, two types of strains exist depending on whether they express a class I or class II pilin[9,10]. These two strain types are nearly equally represented in clinical isolates[10]. In class I pili expressing strains, a first level of variation is linked to a process often refered to as *antigenic variation* during which bacteria change the pilin amino acid sequence via DNA recombination. As the changes in sequences mostly occur in a carboxy terminal region exposed to the outer environment, such region is termed the hypervariable loop. Because of the high frequency of this process, a bacterium of a given class I strain can evolve its pilin sequence during an infection process[11]. In contrast, in class II pilin expressing strains, pilin sequence remains constant with limited variation between strains.

A second level of variation is linked to posttranslational modifications (PTM). In class I strains, a hexose is positioned on serine 63. In class II strains, glycosylation can occur at several sites, the number and glycosylation site being variable in different strains[12]. The nature of the hexose on serine 63 of class I pilins is defined by the presence of the *pglB1* or *pglB2* allele on the genome of the strain of interest which will lead to the expression of a diacetamido-tri-deoxy-hexose (DATDH) or a glyceramido-tri-deoxy-hexose (GATDH) respectively[13,14]. A second or even a third hexose can then be added sequentially to the first G/DATDH by PglA and PglE glycosyl transferases respectively[15]. This is another source of variation as the *pglA* and *pglE* genes are submitted to a genetic regulation process called *phase variation*[8]. This process is mediated by polynucleotide repeat expansion or contraction during replication leading to reversible changes in the reading frame of the gene and thus protein expression.

Finally, a third level of variation comes from another PTM on serine 69, phosphoforms that can be phosphocholine (PC) or phosphoethanolamine (PEA) if the *pptA* allele is present in the strain[16], or a phosphoglycerol (glycerol-3-phosphate or G3P) if the *pptB* allele is present[17]. Immune evasion against antibodies targeting type IV pili is thus attributed to the structural diversity present among strains, including variations in pilin sequence, sugar type, and phosphoform. For a specific strain, mechanisms such as pilin *antigenic* and *phase variation* enable rapid adaptations, further contributing to immune evasion.

The goal of this study was to explore the structural diversity of *Neisseria meningitidis* type IV pili and to better understand how this diversity impacts the immune response by altering antibody binding. Although the overall pilus structure of the *N. meningitidis* type IV pilus has been determined[18] as well as the genetic and chemical sources of surface structure variation have been described, the structural determinants of antigenic variation and immune escape remain unclear. Here, we take advantage of cryo-electron microscopy (cryo-EM) single particle analysis (SPA) and the helical nature of pili to bring a structural understanding of this key immune evasion process against this large polymeric structure. We first determine the structures of two pilin sequence type variants in a class I pilin harboring strain to evaluate the structural impact of variation at the pilin amino acid level in these strains. We then generate and characterize camelid single domain nanobodies directed against type IV pili both as tools to provide

structural information of potential epitopes and as starting points for therapeutic discovery. The structures of two nanobodies complexed with different type IV pilin variants at the PTM level are determined, providing structural insight into pilin surface variation-based immune evasion. Importantly, our work demonstrates that structural information for type IV pili can be harnessed for the rational design of nanobodies targeting a broader spectrum of pilin variants, paving way for the discovery of novel therapeutics against pathogenic *Neisseria* species.

## Results

### Structure of type IV pili PTMs and sequence types

To better understand the structural bases of the variation in type IV pili surface, i.e. PTM and sequence variation, we determined the native structure of type IV pili purified from two *N. meningitidis* sequence types, SB and SA (Supplementary Fig. 1b), of the 8013 strain genetically modified to have reduced antigenic variation[19,20]. This strain expresses a class I pilin of the SB type, a GATDH sugar on S63 and G3P phosphoform on S69.

Purified SB-sequence type IV pili (Supplementary Fig. 1c) tend to bundle (Fig. 1a) but isolated segments were subjected to several rounds of 2D classification to yield centered, vertically-aligned filaments with well-resolved secondary structure features and visible PilE monomers. The averaged power spectra of the particles also allowed to estimate initial helical twist and rise parameters. The final cryo-EM map of pili (Fig. 1b, left, SB sequence type, GATDH, G3P) at an average 2.9 Å resolution was generated from iterative helical refinements (Supplementary Fig. 2a, b). Resolution was homogeneous through the structure (Supplementary Fig. 2c). As previously observed, pili follow 3 types of helical geometries (1-start right-handed, 3-start left-handed and 4-start right-handed)[18]. The twist and rise values for all cryo-EM structures can be consulted in Supplementary Fig. 2d. The density was sufficient to identify the locations and positioning of GATDH (Fig. 1c, yellow) and glycerol-3-phosphate (G3P, red), as well as for building atomic models of monomers or multimers of PilE. Molecular dynamics of the whole pilus with GATDH and G3P for a period of 1μs showed the overall stability of the structure (Supplementary Movie 1, Supplementary Fig. 3a). As previously described, the PilE monomer contains a 4 strand beta-sheet region (Fig. 1c), and an N-terminal melted alpha helix that forms the core of the type-IV pilus[18]. Molecular dynamics of the beta-sheet head in the context of the whole pilus shows that the protruding C-terminal hypervariable region is mobile relative to the rest of the protein occasionally completely detaching from the beta sheet (Supplementary Movie 2, Supplementary Fig. 3b).

The phosphoglycerol appears clearly on S69 pointing away from the structure (Fig. 1c). Molecular dynamics simulations suggest that the glycerol portion of the molecule undergoes rotational motion, which is supported by the lower resolution observed in its electron microscopy (EM) density compared to the phosphate atom (Supplementary Movie 3). Depending on the orientation that the glycerol group adopts, we predict there are occasional interactions with Oγ of S76 and Oε1 of Q78/Q90 that might transiently stabilize the G3P molecule. The GATDH is well resolved (Fig. 1c) confirming its structure initially deduced from fragments analyzed by mass spectrometry[14] with the acetamido and glyceramido moieties on carbon 2 and 4 of the ring. Interestingly, the proximity and relative positioning of GATDH to E56 suggests the presence of a hydrogen $O - H \cdots O$ bond (Fig. 1c). Accordingly, molecular dynamics shows that the allowed C8-C9-C10 bond rotations in GATDH allows O5 at the tip of the sugar to rotate, which in some cases permits PilE to alternate between one and two simultaneous hydrogen bonds between E56 and the sugar (Supplementary Movie 4). Analysis of the dynamics of hydrogen bonds formed between GATDH and E56 side chains over a 1μs period shows that they are present 42% of the time (SD = 18%, average of 30 pilin monomers). This further confirms that the GATDH is mostly immobilized with

 

limited rotation, and might furthermore apply to most strains as E56 is highly conserved in pilin sequences[10].

The structure of pili purified from the 8013 strain expressing the SA sequence type was determined in the same fashion to shed light on the impact of sequence variation on the pilus structure (Fig. 1b, center). Pilin expressing the SA and SB sequence types have the same PTMs but they differ in their amino acid sequences (Supplementary Fig. 1b). Comparison of pili from strains expressing pilin of SA and SB sequence types reveals nearly identical backbone structures except in the C-terminal hypervariable loop (Fig.1d). In this region, the SB structure shows a longer and more protrusive extension, mainly due to aminoacidic deletions in SA. Accordingly, molecular dynamics analysis of the SA pilin shows limited movement in the hypervariable region compared to the SB sequence type (Supplementary Fig 3b). Since mutations of charged amino acids in this area of the SB sequence caused loss of bacterial aggregation[21], the global change of shape of

this loop between the SA and SB sequence types is likely to affect the basis for pilus-pilus adhesion. Interestingly, although the backbone structure is identical, several side chains in the first strand of the beta sheet, close to the alpha-beta loop, are also different between sequence types (Fig.1d).

In summary, our structural analysis demonstrates that the combination of sequence variation, phosphoglycerol moiety, and the GATDH sugar covers a significant portion of the pilus surface, which can exhibit changes within a specific strain or among different strains (Fig. 1b, right).

## Type IV pili directed nanobodies recognize posttranslational modification and amino acid sequence

In order to determine how antibodies are generated against this complex surface structure and how *N. meningitidis* escapes this response we sought to isolate antibodies against type IV pili and

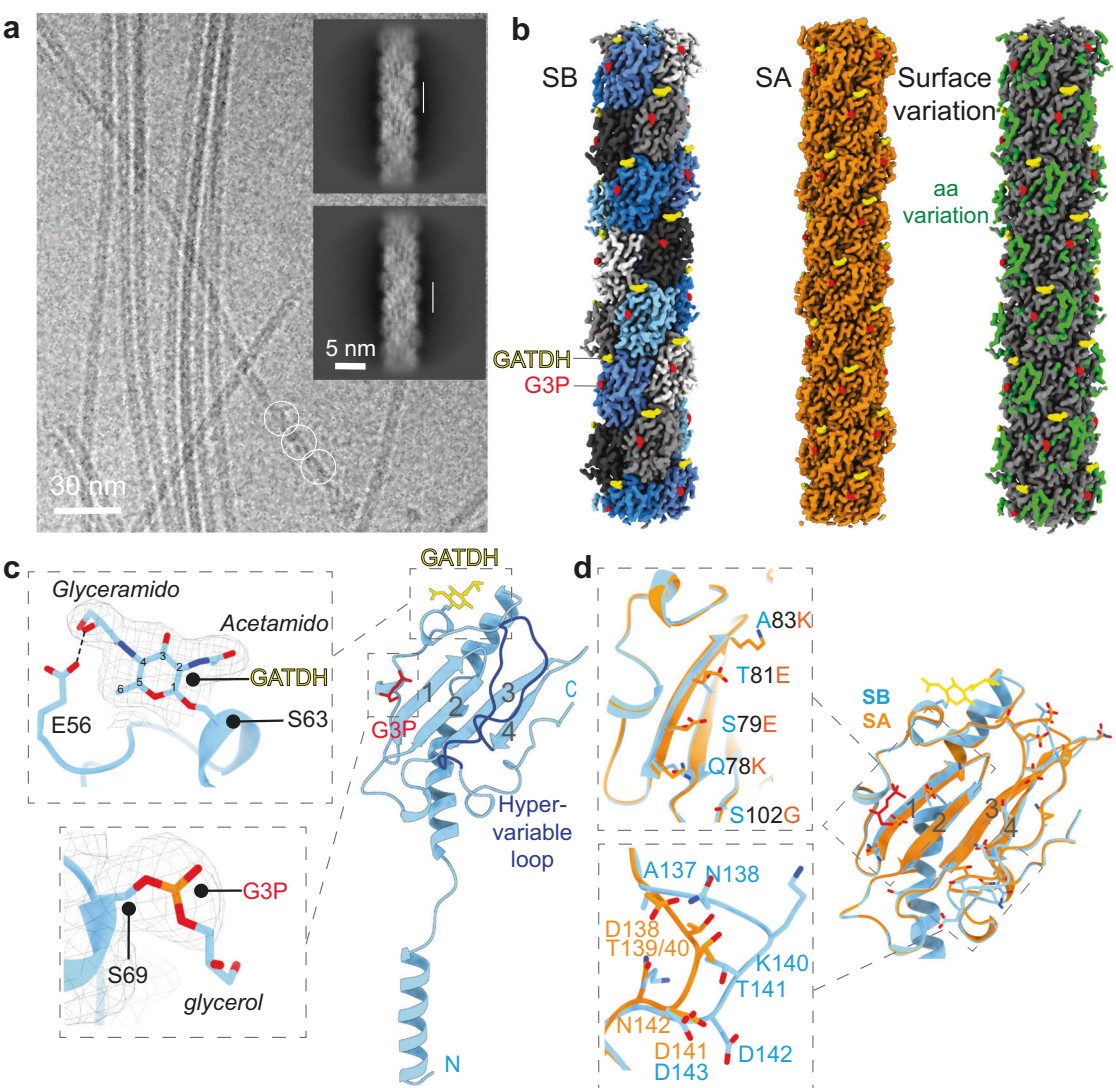

**Fig. 1 | CryoEM structure of pili surface variation. a** Illustrative Cryo-EM micrograph of type IV pili (SB sequence type, GATDH, G3P). Filament segments were selected (white circles). 2D classes averages (inset) already show secondary structure features that allow to identify monomers (vertical white bars). **b** CryoEM maps of SB and SA sequence types; *left*, cryoEM map at 2.99 Å resolution of the SB sequence type pilus, pilin monomers appear in different colors, G3P in red and GATDH in yellow; *middle*, cryoEM map at 3.15 Å resolution of the pilus made of the SA sequence type pilin (GATDH, G3P); *right*, pilus surface showing variable regions, amino acid changes between SA and SB sequence types (green), G3P (red) and

GATDH (yellow). **c** Individual SB pilin subunit with G3P and GATDH, the hypervariable loop appears in dark blue. Close-ups of the phosphoglycerol moiety on S69 and on the GATDH moiety present on S63 are indicated. The dashed line represents a potential hydrogen bond. **d** Overlay of the SB (blue) and SA (orange) sequence type structures. Side chains of amino acids that are different between the two sequence types appear as sticks. Close-ups of regions of interest are presented; top, amino acid changes in strand 1 of the beta sheet head; bottom, comparison of the hypervariable loop of SB and SA sequence types.

determine their structure in complex with type IV pili. Because of their small size and amenability to large scale purification and structure determination we opted for nanobodies, and an alpaca was inoculated with native type IV pili purified from the *N. meningitidis* 8013 strain.

Pili purified from the SB sequence expressing 8013 strain were assessed for purity using SDS-PAGE (Supplementary Fig. 1c) and negative stain electron microscopy (Fig. 2a) prior to immunization. Following nanobody domain amplification from lymphocytes, clones expressing nanobodies reacting with pili were enriched by phage display after type IV pili immobilization on beads. Sequencing of the selected clones revealed six groups of sequences (Supplementary Fig. 4a, b). In each of these groups of related sequences one clone was selected to test binding with type IV pili expressed at the surface of bacteria adhering to endothelial cells (Fig. 2b–d). As a control, binding was also tested with the previously described mouse monoclonal antibody (20D9) which recognizes *N. meningitidis* type IV pili[22]. As expected, using this approach the 20D9 antibody revealed a

meshwork of type IV pili associated with bacterial microcolonies on the surface of cells (Fig. 2b). The 6 nanobodies that were subsequently tested generated different levels of signals with F10 and C24 showing the strongest signal and were selected for further studies.

We then sought to better characterize the binding site of these two nanobodies in relation with the variable pilus structures using bacterial mutants (Fig. 2d). Pili from a *pglD* mutant, unable to synthesize the GATDH hexose on serine 63, were not recognized by either the C24 or the F10 nanobodies pointing to a key participation of glycosylation and this region in the interaction. In this mutant, pili were produced to the same level as the wild-type strain[23]. Absence of G3P modification on serine 69 in a *pptB* mutant did not affect the binding of the F10 nanobody but, in contrast, strongly affected C24 binding showing different binding sites for these nanobodies.

The impact of amino acid variation was then tested using the naturally occurring SA variant of the pilin. Neither the C24 nor the F10 nanobody recognized pili expressing the SA variant of the pilin

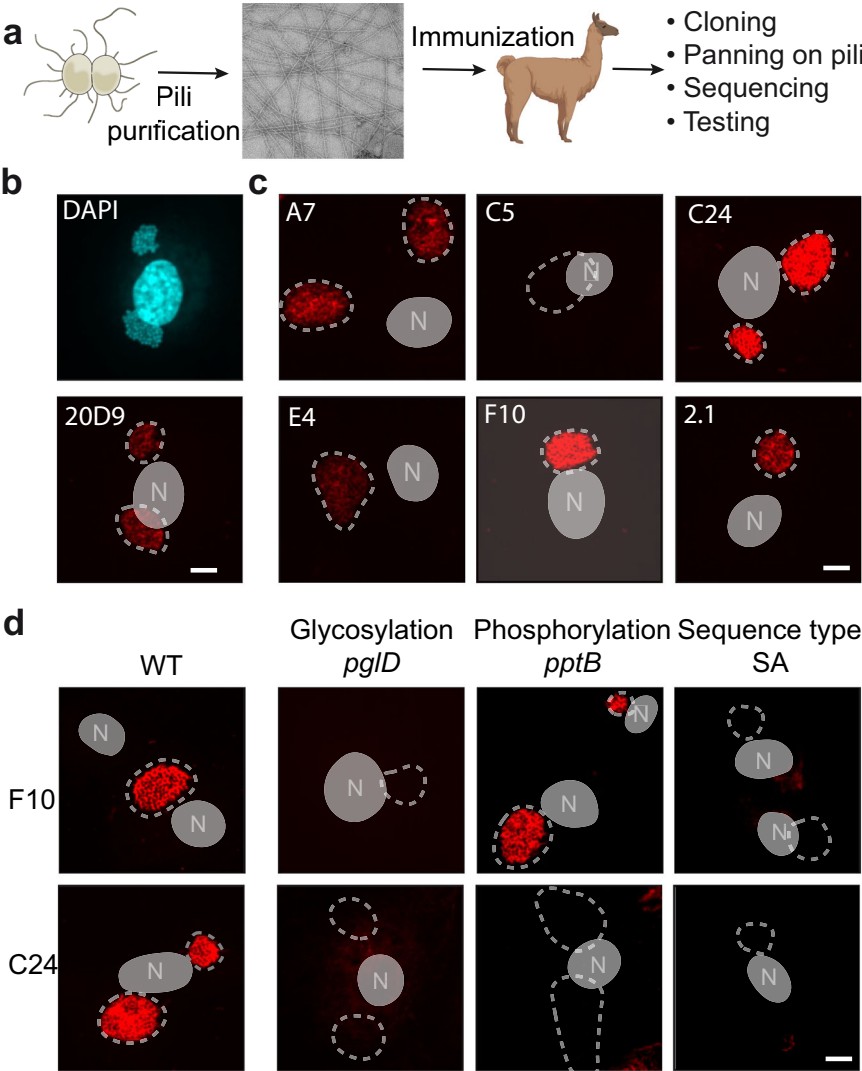

**Fig. 2 | Isolation of a set of nanobodies reacting with type IV pili. a** Illustration of the procedure, pilus isolation, alpaca immunization and nanobody domain cloning. Created with BioRender.com. **b** Labeling of type IV pili expressed on the surface of bacteria adhering to endothelial cells with the 20D9 monoclonal antibody[22]. Top panel, DNA staining labeled in blue shows the cell nucleus and two bacterial aggregates. Bottom panel, labeled pili on the same infected cell appear in red; the position of bacterial aggregates identified with the DNA stain were added with a dotted line and the nucleus with a gray shape with the letter N. The same

representation of bacterial colony and nucleus was used in (**c**) and (**d**). **c** Testing of the ability of 6 representative nanobodies to react with pili. For comparison purposes the same settings were used for pilus image acquisition and display for the different nanobodies. **d** Determination of the recognition site of the F10 and C24 nanobodies using mutants affecting different variable surface structures, glycosylation (*pglD*), phosphorylation (*pptB*) and amino acid sequence (SA). The scale bars on (**b–d**) correspond to 5 μm. Each of the immunofluorescence experiments were performed 3 times.

pointing to a role of the hypervariable region in the pilus-nanobody interactions. These results based on immunofluorescence staining were confirmed by ELISA on immobilized bacteria after fusing nanobodies with a mouse antibody Fc region (Fig. 3a, b). Results thus point to a partly overlapping binding site of the F10 and C24 nanobodies involving the hypervariable loop and the GATDH hexose, while C24 binds to the G3P moiety.

Since the interaction between type IV pili is essential for bacterial auto-aggregation and consequently infection progression, the ability of nanobodies to neutralize this process was tested. Accordingly, nanobodies fully inhibited auto-aggregation at a concentration of 500 ng/well (5 µg/ml), while a control irrelevant nanobody did not have any inhibitory effect (Fig. 3c). To explore potential applications in vivo, the ability of nanobodies to opsonize bacteria in vivo was tested. When introduced in the mouse circulation bacteria are cleared rapidly in particular by the phagocytic Kupffer cells in the liver[24], a process enhanced by opsonization. We reasoned that fusing our nanobodies with the mouse antibody Fc region would promote activation of the immune system and in particular by mononuclear phagocytes. Two- and four-hours post i.v. injection, the number of bacteria in the blood slightly decreased in the control group (Fig. 3d). In the presence of F10 and C24, the decrease in circulating bacteria was much sharper. Type IV pili directed nanobodies were thus able to opsonize bacteria in the circulation and favor bacterial elimination by phagocytes.

We have therefore generated two nanobodies that interact with different surface structures of type IV pili both in vitro and in vivo, providing insights into immune evasion and potential clinical applications.

## Cryo-EM structure of nanobody-type IV pili polymeric complexes

The structures of the F10 and C24 nanobodies complexed with pilus structures were determined by cryoEM to gain structural insight into type IV pilus immune recognition. In both cases, images revealed pili with a rugged surface (Fig. 4a) compared to pili alone indicating that nanobodies covered the entire pilus length. Structures were determined as above with similar twist and rise values as their unbound counterparts indicating that nanobody binding did not affect overall pilus structure (Fig. 4b, c). Maps reveal nanobodies bound to the entire surface of the pilus with a 1:1 stoichiometry with pilin monomers. In accordance with the results obtained above (Fig. 2d), the F10 nanobody binds alongside the pilin face with the interaction surface spanning the PTM containing alpha-beta loop and the hypervariable region (Fig. 4d). In contrast, the C24 nanobody interacts mostly with the alpha-beta loop (Fig. 4e) and does not rely on the hypervariable loop but is nevertheless likely interacting with a region containing amino acid changes between the SA and SB sequence types. The interacting surfaces of the two nanobodies on the pilin thus have a small overlap but bind to distinct areas of the pilin surface (Fig. 4f).

In the case of F10, the nanobody complementarity-determining region 1 (CDR1) does not seem to contribute significantly to the nanobody binding, it remains free, away from the pilus structure. CDR2 loop residues [S54-G58] contact PilE_S62 and GATDH on PilE_S63, with many close contacts and a few hydrogen bonds established (Supplementary Fig. 5a). The GATDH moiety is buried in a cleft between PilE and the nanobody (Fig. 5a). A hydrogen bond likely links F10_S55 and GATDH (Fig. 5b). Concerning the CDR3 loop, residues [V102-G114] contact part of the hypervariable loop of PilE, R126-A129, plus N138 further down on the same loop (Fig. 5c, d). The shape complementarity between the surfaces is highly precise, exhibiting a buried surface area of 659.7 Å². In agreement with the binding assays mentioned above (Fig. 2d), the F10 interaction surface on the pilin is away from the phosphoglycerol moiety on the pilin. This explains why F10 binding is independent from the presence of the phosphoform.

The C24 nanobody has an overlapping but clearly distinct binding area, mostly on the alpha-beta loop and away from the hypervariable loop. In addition, unlike F10, C24 binding to PilE occurs with its CDR1 and CDR3 loops. The CDR3 loop residues [R103-V110] compose most

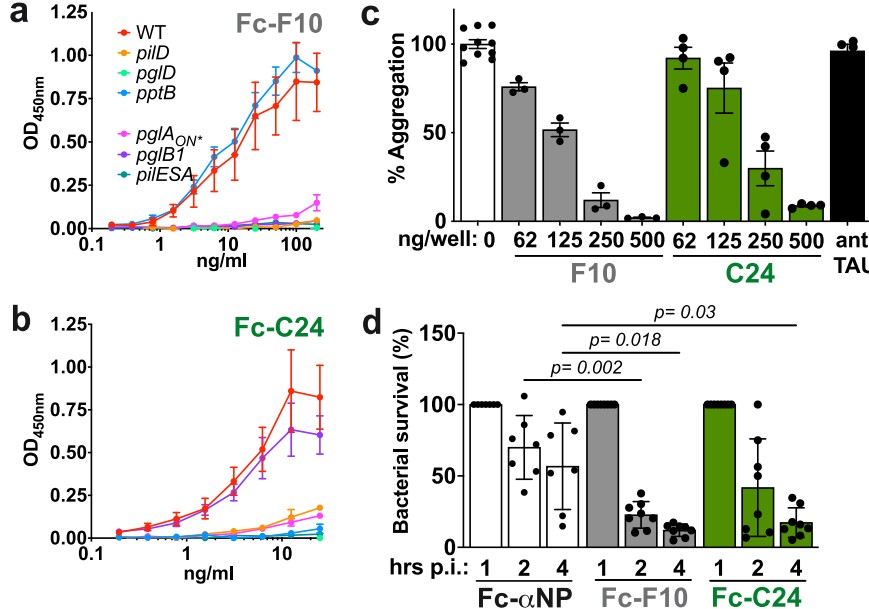

**Fig. 3 | Characterization of the F10 and C24 nanobodies. a** ELISA assay using the F10-mFc fusion on the wild type and indicated strain immobilized on the bottom of the ELISA plate. **b** Same as in A using the C24-mFc fusion. For (**a, b**), graphs represent the result of 3 independent experiments each done in triplicate. Data represent the mean ± SEM. **c** Impact of the indicated amount of F10 (gray) or C24 (green) on the ability of meningococci to form type IV pili-dependent aggregates. A nanobody directed against the Tau protein was used as a negative control[36] at a concentration of 5 µg/ml (black). Results from 3 independent experiments each done in triplicate are presented. Data represent the mean ± SEM. **d** Impact of nanobodies fused to mouse antibody Fc regions on the survival of bacteria in the circulation. Fc-fused nanobodies were injected at 1 h post-infection and the bacteria in the blood collected at indicted times. A nanobody directed against the SARS-Cov2 nucleoprotein (Fc-αNP) was used as a control[34]. A total of 8 mice per group were used in two independent experiments. Data represent the mean ± SEM. After checking for normality using a Shapiro-Wilkinson test, a two-way ANOVA statistical test was performed. Source data are provided as a Source Data file.

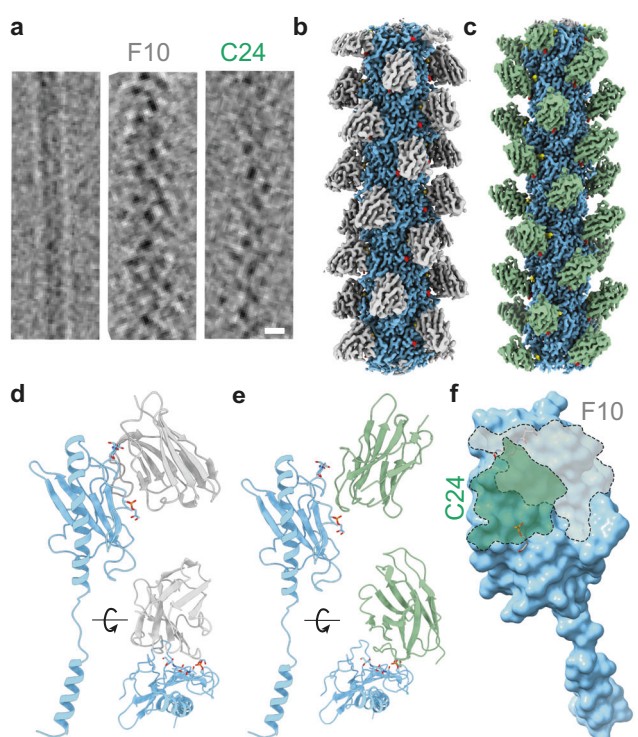

**Fig. 4 | Overall structure of F10 and C24 nanobodies complexed with the pilus (SB sequence type). a** Illustrative Cryo-EM image of type IV pili alone or complexed with the F10 and C24 nanobodies. Scale bar corresponds to 5 nm. **b** Color coded final Cryo-EM map of the SB-GATDH pilus structure (blue) complexed with the F10 nanobody (gray) at 2.92 Å resolution. **c** Cryo-EM map of the pilus structure (blue) complexed with the C24 nanobody (green) at 2.90 Å resolution. **d** Ribbon structure of the F10 nanobody complexed with the pilin monomer. **e** Ribbon structure of the C24 nanobody complexed with the pilin monomer. **f** Binding sites of the F10 and C24 nanobodies on the pilin surface.

of the contact surface and hydrogen bonds, with a minor contribution at the basal section of R30 and Y33 on the CDR2 loop (Supplementary Fig 5b). Shape complementarity is also pronounced with a buried surface area of 510.1 Å$^2$.

Interestingly, the C24-GATDH binding cleft is much more open compared to F10 (Fig. 5e). C24 R103 forms a hydrogen bond with the acetamido arm of the GATDH (Fig. 5f). The phosphoglycerol modification is located in a cleft at the border of the interaction region, glycerol part of the PTM is sticking out (Fig. 5g). The phosphate head forms H-bonds with Oγ of S105 and the backbone nitrogen from the main chain of R106 (Fig. 5h). The glycerol part of the modification does not seem to tightly interact with the nanobody.

The anti-T4P nanobodies exhibit distinct binding to various regions on the pilin monomer, predominantly involving variable regions. F10 primarily interacts with the amino acid sequence, while both F10 and C24 engage with the GATDH moiety, and C24 additionally interacts with G3P. As a result, any surface structure variation is likely to hinder the binding of the nanobodies (Fig. 4f).

### Impact of type IV pili surface variation on nanobody binding

We then sought to better understand how surface pilus variation affect the antibody response by testing the binding of F10 and C24 nanobodies to different pilus variants. As shown above, recombination-based amino acid changes had a strong impact on nanobody binding and neither F10 or C24 showed any binding to SA sequence type pili (Fig. 3a, b). In the case of F10 this is easily understandable since the nanobody epitope largely encompasses the C-terminal hypervariable loop (Fig. 4d). For instance, PilE-SA_K136 clashes with F10_T109 and

PilE-SA_K83 clashes with F-10_I105 (Fig. 6a, b). In the case of C24 this is more surprising since this nanobody binds primarily on the alpha-beta loop which is identical between SA and SB sequences. Structures reveal that clashes come from amino acid differences located in beta sheet strand 1. PilE-SA clashes with C24 in several instances: SA_K83 with C24_S108, SA_E81 with C24_S105, and a clash of SA_G3P with C24_R106 and C24_L107 likely due to a forced outwards repulsion of the glycerol group by SA_E79 (Fig. 6c, d).

The impact of glycosylation variants was then explored. Certain strains display a DATDH rather than a GATDH on serine 63. These two sugars are very similar and differ only by one modification of the ring which in one case is an acetamido (DATDH, *pglB1*) and in the other a glyceramido (GATDH, *pglB2*) moiety. To address the impact of the presence of DATDH on antibody binding, a strain expressing the *pglB1* gene rather than *pglB2*, and thus displaying a DATDH modification, was generated and the structure of pili determined from this strain (Fig. 7a). The presence of DATDH rather than GATDH was verified by mass spectrometry (Supplementary Fig. 6). The structure also revealed the two symmetrical acetamido groups on positions 2 and 4 without any contact with E56 (Fig. 7b).

C24 interaction with pili was not affected by the presence of DATDH rather than GATDH as shown using an ELISA assay (Fig. 3b), or by immunofluorescence staining (Fig. 7c). The structure of C24 complexed with the pilus expressing the DATDH moiety was determined revealing that binding of C24 to DATDH was also stabilized by R103 in the nanobody (Fig. 7d and Supplementary Fig 5c). In contrast, despite the minor difference between the two sugar forms, the presence of a DATDH had a strong impact on F10 binding in both assays (Fig. 3a and Fig. 7c). The reason for the absence of binding of F10 could be due to several reasons either independently or in combination: (i) DATDH is not stabilized through the formation of a hydrogen bond with E56 of PilE; (ii) the buried surface area is smaller; (iii) less hydrogen bonds are formed between the sugar and F10. In any case this shows the importance of this seemingly minor modification as an immune escape mechanism.

Another source of variation in the sugar moiety is due to the addition of a second sugar, a galactose in position 3 of the D/GATDH[25]. In this case, neither F10 nor C24 could bind to pili displayed by strains expressing the *pglA* transferase gene in the ON phase (Fig. 3a, b). In the case of F10, this is easily explained by the fact that carbon 3 of GATDH is deeply buried in the structure (Fig. 5a). In the case of C24 this is likely due to the position of the galactose, almost perpendicular to GATDH which generates clashes with S112 and S114 in C24 (Fig. 7e). Accordingly, introducing a double mutation in the C24 nanobody (S112G and S114G) showed increased binding to pili with the *pglA*$_{ON*}$ strain in the immunofluorescence assay (Fig. 7f). Conversely, the C24_S112-114G nanobody displayed an apparent lower affinity compared to the reference C24 nanobody (Fig. 7g). These qualitative results were confirmed by ELISA (Fig. 7h, i).

### Discussion

The main objective of this study was to gain a structural understanding of T4P-based bacterial immune evasion, with the ultimate aim of developing diagnostic and therapeutic tools. In pursuit of this goal, we elucidated the structures of six pili complexes using cryoEM. Our findings unveiled the extensive alterations occurring across the entire exposed pilin surface, providing valuable insights into the remarkable immune evasion mechanisms of pathogenic bacteria. Beyond changing its amino acid sequence, the largely unfolded hypervariable loop can also display extensive movements that could impact antibody binding by limiting the availability of specific epitopes. This is particularly true for the examples of the SB sequence type which is longer than the SA sequence type loop. The situation for the alpha-beta loop is different since it is conserved at the amino acid level, but it is highly modified with PTMs, glycosylation and phosphoform, which can

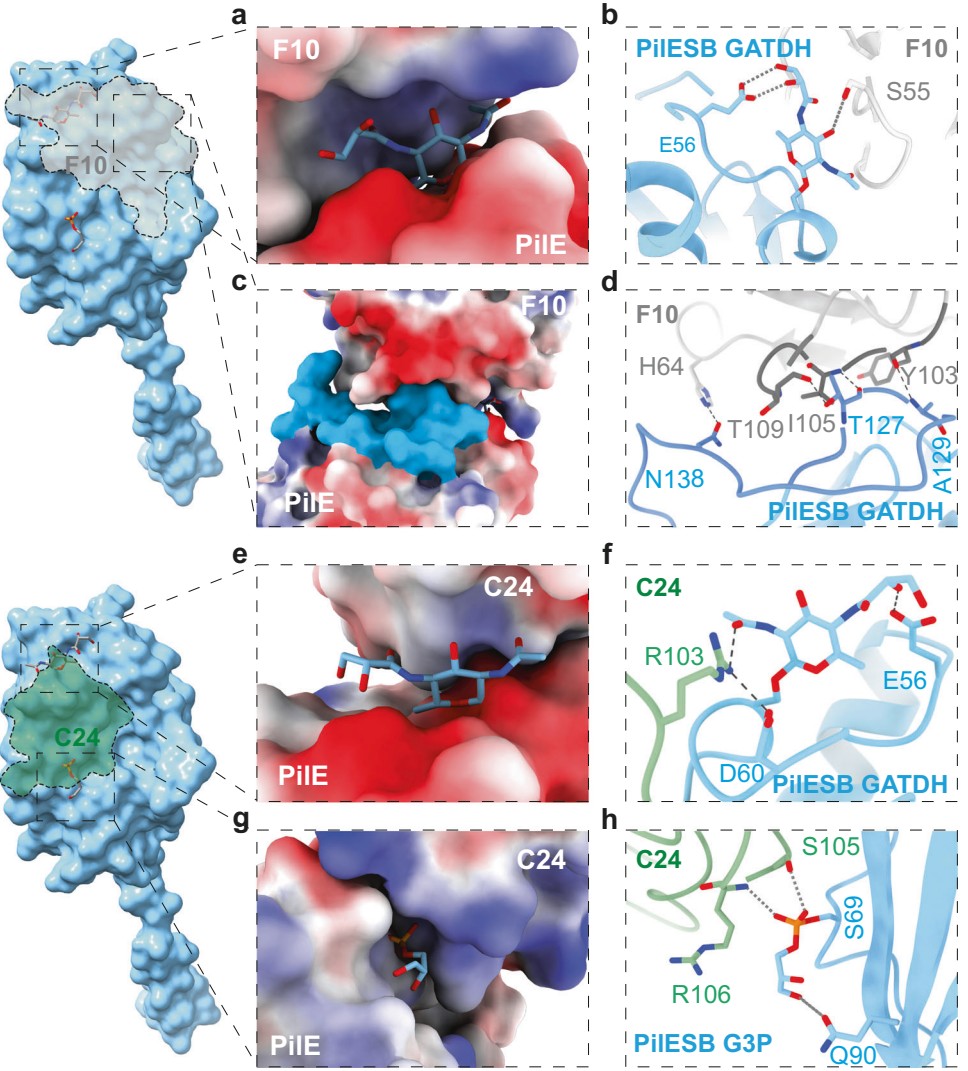

**Fig. 5 | Detailed structures of the interaction of nanobodies with post-translational modifications and amino acid sequence variants. a** Surface representation of F10 complexed with PilE with GATDH represented as sticks. **b** Interaction between F10 and the GATDH. **c** Surface representation of the F10 nanobody with the pilin. The hypervariable loop is indicated in blue. **d** Interactions between F10 and the PilE hypervariable loop. **e** Surface representation of C24 complexed with PilE with GATDH represented as sticks. **f** Interaction between C24 and the GATDH moiety. **g** Surface representation of C24 complexed with PilE with the phosphoglycerol represented as sticks. **h** Interactions between C24 and the the phosphoglycerol.

change between different strains as well as in a given strain through cycles of divisions. Molecular dynamics also reveal the extent of the movement of the PTMs on the surface making the surface less ordered. Furthermore, strand β1 of the beta sheet (Supplementary Fig. 1B) undergoes several amino acid changes upon switching between sequence types leading to alterations of the pilin surface close to the alpha-beta loop where the PTMs lie. We have also shown previously that in certain conditions a phosphoform can be added to S93 which adds another layer of surface change[17].

Overall, the bacterium presents a pilus surface which seems entirely variable with extensive movements of surface structures providing an impressive toolkit for immune evasion. This is particularly striking since type IV pili are multifunctional virulence factors and their structure should be under tight control to maintain structure and function[3,21]. Previous studies have shown that pilin surface changes have surprisingly little effects on activities like adhesion or transformation[12]. Nevertheless, considering that auto-aggregation is driven by pilus-pilus interactions, the important differences in the surface of the SA and SB sequence type pili likely explain their different aggregation phenotypes. Further addition of a phosphoform on S93

also leads to less aggregation[17] also pointing to the importance of the pilin surface for auto-aggregation.

To explore how antibodies interact with this complex and changing pilus surface we produced anti-T4P nanobodies. Camelid nanobodies were selected, over mammalian antibodies, first because of their amenability to large scale purification and structural analysis. Another motivation to use nanobodies was the hope to identify small, conserved surfaces onto which these small proteins could fit. The interaction of nanobodies all along the pilus length and the helicoidal nature of type IV pili allowed us to determine the structure of the complex at high resolution. The structure of the F10-pilus complex explains the high specificity of this nanobody for the SB sequence type over the SA sequence type because its binding site is located on the side of the pilin surface containing the hypervariable loop. In contrast, the F10 binding site on the pilus surface remains at a distance from the G3P explaining why the absence of this PTM does not alter F10 binding. Unexpectedly, by analyzing the T4P-C24 cryoEM structures, we also found that the binding of a nanobody targeting the alpha-beta loop could be altered by amino acid changes in strand β1 of the beta-sheet region. Long side chain amino acids from the beta sheet come in close

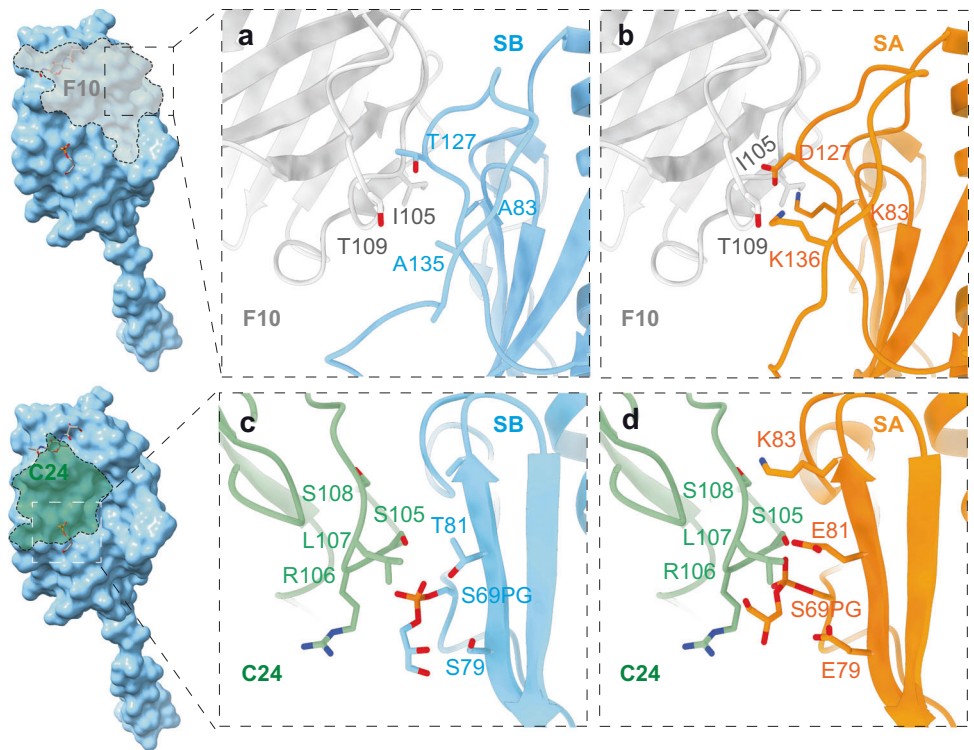

**Fig. 6 | Impact of amino acid changes on nanobody binding. a** Close-up of the F10/pilin binding region involving the hypervariable loop of the SB sequence type pilin (PilE aa 120-154). **b** Identified amino acid clashes of SA-variant mutations in the hypervariable loop preventing the binding of the F10 nanobody. **c** Close-up of the C24/SB pilin binding region involving the β1 pilin beta-sheet strand of the SB sequence type and the phosphoglycerol PTM. **d** Clashes preventing C24 binding to the SA sequence type.

contact with the alpha-beta loop and perturb binding. Small changes in sugar structure also have deep effects on nanobody binding. The mere difference of one carbon atom between GATDH and DATDH significantly impacts antibody recognition, as observed in our detailed investigation of one of our nanobodies (F10). Even the phosphoforms can have a strong effect on binding as in the case of the C24 nanobody which depends on the G3P for efficient binding. Our study demonstrates that all surface modifications can partially reduce or fully abrogate nanobody binding. Importantly, we offer a structural explanation for these observations.

It should also be pointed out that our study has focused on a class I pilin which represents about 50% of *N. meningitidis* strains[10]. The other half of pathogenic strains express a class II strain which do not have the ability to change their amino acid sequences but compensate this by having several glycosylation sites covering the pilin surface[12]. How antibodies recognize such pilin surfaces remains to be determined but this study provides a broad framework that can be used for class II pilins as well as pili from other species.

Our study also shows the limits of the bacterial immune evasion process and opens strategies to bypass them. Our results show that nanobodies can be used to detect pili with various approaches, ELISA and immunofluorescence. Furthermore, nanobodies can be used to perturb type IV functions, auto-aggregation in particular. Our results are also encouraging in terms of therapeutics since functionalization of nanobodies with an Fc fragment allows the innate immune system to eliminate bacteria from the blood stream. The exact mechanism of this clearance will need to be further studied. It could be linked to opsonization dependent complement activation or to enhanced phagocytosis, in particular by Kupffer cells in the liver.

The structures we describe also provides insights to generate broadly covering tools. For instance, structures reveal a small region

between two pilin monomers which bears little surface variation. This region covers the apical section on PilE (top part of the hypervariable loop and the sugar) and loop 92-102 of neighboring PilE monomers from the quaternary structure. Alternatively, a combination of nanobodies such as the ones we describe here could be used. Although the exposed pilin surface is small, different nanobodies can bind to specific areas of the pilus structure. This suggests that a panel of nanobodies could be established to bind to a broad range of pilus variants. In addition, certain nanobodies can individually bind different variants. As explained above, the F10 nanobody for instance binds to the pilus structure independently of the phosphoform. Although the C24 nanobody is in tight interaction with the sugar moiety it can bind to both GATDH or DATDH with the same efficiency. The presence of a second sugar on the GATDH was shown to perturb binding, but the availability of the structures allowed to rationally design a variant of C24 with a higher affinity of the GATDH-GAL variant. The structures presented in this study serve as a foundation for numerous further investigations to generate therapeutically useful nanobodies. The combination of cryoEM and molecular dynamics used here can be used to enhance the affinity between pilin and nanobodies, as well as to expand the binding capacity to various pilin variants. While identifying a universally binding single nanobody might prove challenging, finding a small set of nanobodies that can respond to the different PilE surface regions identified might be sufficient to enable treatment and/or diagnosis. The design of multi-specific nanobody or nanobody-like constructs with Fc-mimicking domains is indeed possible, as evidenced by recent developments in the drug-design field such as TriTACs[26] or DARPins[27]. Short, synthetic peptide-mimetics based on the CDR sequences can also be explored[28]. Overall, our work opens new avenues of research and show promise to use type IV pili as diagnostic and therapeutic tools for these deadly infections.

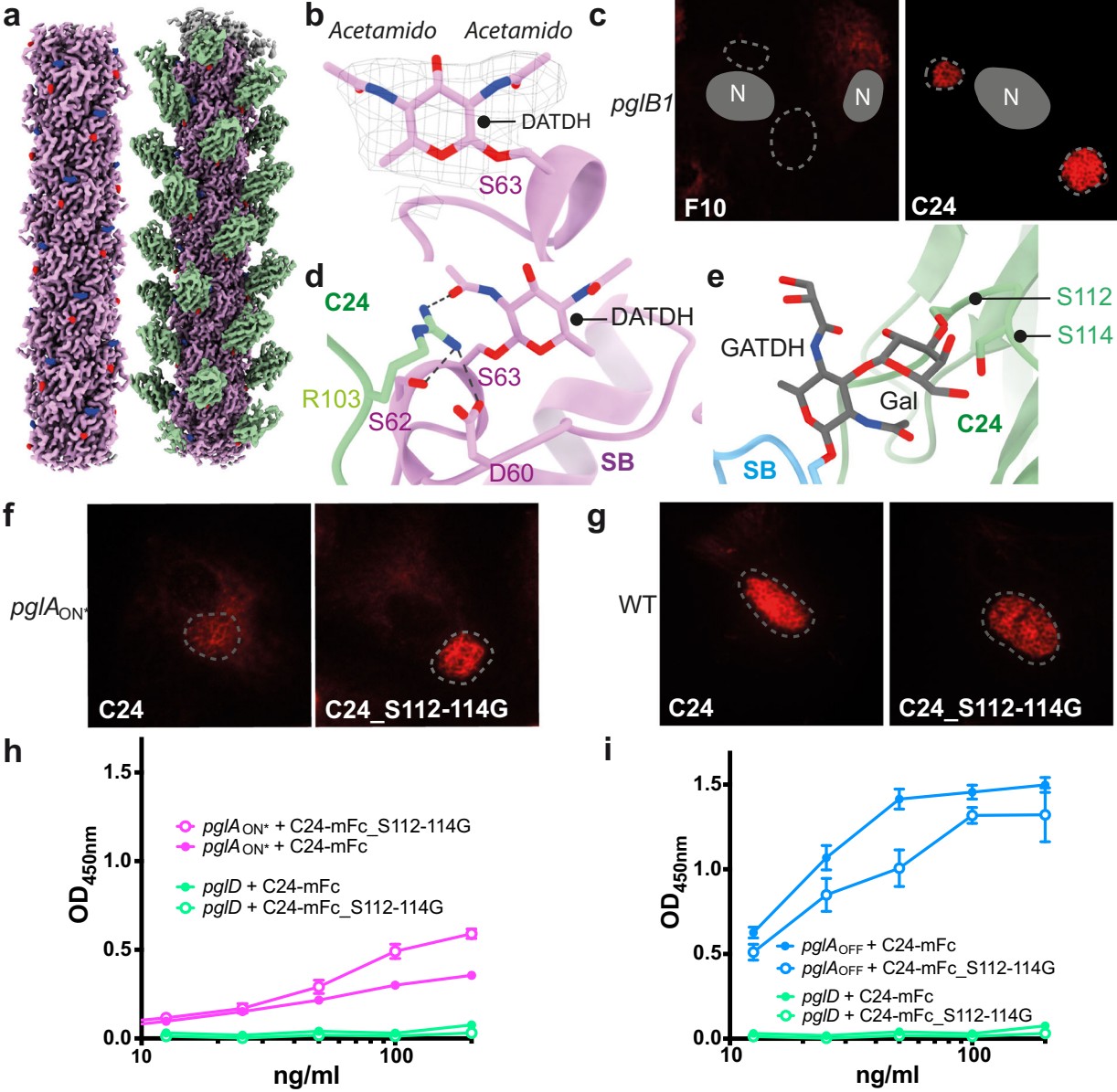

**Fig. 7 | Impact of sugar variants on nanobody binding. a** Sharpened cryo-EM map of SB_DATDH (left) and the same structure complexed with the C24 nanobody (right). The PilE monomer is pink, DATDH is blue and G3P is red. **b** Close up of DATDH within the EM density. **c** Nanobody staining of HUVECs infected with the *pglB1* strain displaying DATDH, with F10 (left) and C24 (right). **d** Close-up of PilE and DATDH interactions with C24_R103. **e** The GATDH-aligned double sugar (GATDH-GLA) attached to S63 modeled in the SB-GATDH-C24 structure reveals clashes with S112 and S114. **f** Type IV pili detection on HUVEC cells infected with the *pglA*ON* strain expressing GATDH-GLA double sugar with the C24 and C24_S112-114G nanobodies. **g** Type IV pili detection on HUVEC cells infected with the GATDH

expressing wild type strain. The same parameters were used for image acquisition and display. For (**c, f, g**), immunofluorence assays were done in 3 independent experiments. **h** ELISA assay characterization of the C24-mFc_S112-114G mutant binding to the *pglA*ON* strain which expresses the SB sequence type with a GATDH-Gal glycosylation. **i** ELISA assay characterization of the C24-mFc_S112-114G mutant binding to the *pglA*OFF strain which, as the WT strain, expresses the SB sequence type with a GATDH. For (**h, i**), graphs represent the result of 3 independent experiments each done in triplicate. Data represent the mean ± SEM. Source data are provided as a Source Data file.

## Methods

### Antibodies and chemicals
The following antibodies were used for immunofluorescence: mouse monoclonal antibody anti-PilE, clone 20D9 (1 μg/ml, Pujol et al., 1999) and mouse anti-His tag (3.5 μg/ml, Biolegend, #BLE906102). The following goat secondary antibodies were used for immunofluorescence and enzyme-linked immunosorbent assay (ELISA): anti-mouse IgG (H + L) coupled to Alexa Fluor 488 or 568 (1/200, Invitrogen, #A21121 and #A11004), F(ab')₂ Fragment Goat Anti-Mouse IgG Fcγ fragment coupled to horseradish peroxidase (1/2500, Jackson ImmunoResearch, #115-036-071). 4′,6-diamidino-2-phenylindole (DAPI, 0.5 μg/ml) was

purchased from Thermo Scientific™, #62247. Trypsin-EDTA (0.05%) was purchased from Gibco. TMB Substrate Reagent for ELISA was purchased from BD BIOSCIENCES, #555214. Isopropyl ß-D-1- thiogalactopyranoside (IPTG, 0.5 mM) was purchased from Euromedex, #EU0008-B. Polymyxin B Sulfate (1 mg/ml) from Sigma-Aldrich, # 5291-1GM. High Density Cobalt Agarose from Interchim, #6BCL-QHCO-25.

### Bacterial strains and growth conditions
Genetically modified *N. meningitidis* strains used in this study (Supplementary Table 2) were derived from the 8013 /clone 12 (2C43) strain

expressing the SB pilin variant[19]. Bacteria were grown on Gonococcal Medium Base (GCB, BD Difco) agar plates supplemented with Kellogg's supplements[29] and, when required, 100 µg/ml kanamycin, 50 µg/ml spectinomycin, or 5 µg/ml chloramphenicol, at 37 °C in moist atmosphere containing 5% $CO_2$. Before adhesion and aggregation assays, bacteria were grown with shaking at 37 °C and 5% $CO_2$. Bacteria were inoculated at an $OD_{600}$ of 0.05 in Human Endothelial-SFM or RPMI (Gibco) supplemented with 10% heat-inactivated fetal bovine serum (FBS) and grown for 2 h-2 h 30. A strain containing a deletion of the guanine quartet upstream of the *pilE* gene (Kennouche et al., [21]) was used to minimize antigenic variation. Certain strains are described elsewhere, *pglD*[30], *pilD*[30], *pptB* (Chamot-Rooke et al. [17]) and the SA pilin sequence variant (Nassif et al. [19]). *Escherichia coli* transformants (XL1-Blue or BL21 (DE3)/pLys) were grown at 37 °C in liquid or solid Luria-Bertani medium (BD Difco) containing 50 µg/ml kanamycin.

## Cloning and mutagenesis

**pglB1 expression in the 8013 strain.** The Gibson assembly strategy was used to substitute the *pglB1* (DATDH) gene from the *Neisseria gonorrhoeae* MS11 strain into the *N. meningitidis* ΔG4 strain (GATDH). The *pglB1* ORF without the first 23 bp was amplified with Phusion Plus DNA Polymerase (Thermo Scientific, #F630S) from MS11 chromosomal DNA using the primers *pglB1*_MS11_fwd and *pglB1*_MS11_rev (Supplementary Table 2). A second PCR fragment containing a $Kan^r$ cassette was obtained from plasmid pT1K1[31] with primers Km_*pglB1*_fwd and Km_*pglB1*_rev. A third PCR fragment contained the downstream region (*HAD*) from *pglB2* of *N. meningitidis* 8013 was prepared with HAD_*pglC*_fwd and HAD_*pglC*_rev primers. The three PCR fragments, plus XhoI and SacI-linearized pBlueScript SK(+) vector (Stratagene), were purified and then assembled with the Gibson Assembly Master Mix (New England BioLabs). The plasmid, containing the $Kan^r$ marker between *pglB1* and *HAD*, was isolated from *E. coli*, sequenced, and used to transform the ΔG4 strain.

**Stable expression of the pglA gene in the 8013 strain.** To introduce the *pglA*$_{ON}$ gene into ΔG4, a part of the *pglA* ORF, encompassing polyG tract was amplified from MC58 (menB *N. meningitidis* strain) chromosomal DNA using the primers *pglA*_MC58_fwd and *pglA*_MC58_rev. A second PCR fragment containing a $Kan^r$ cassette was obtained from plasmid pT1K1 with primers Km_*pglA*_fwd and Km_*pglA*_rev. A third PCR fragment contained the downstream region from *pglA*$_{OFF}$ of *N. meningitidis* 8013 was prepared with *pglA*_2C43_fwd and *pglA*_2C43_rev primers. The three PCR fragments, plus XhoI and SacI-linearized pBlueScript SK(+) vector (Stratagene), were purified and then assembled with the Gibson Assembly Master Mix (New England BioLabs). The plasmid was isolated from *E. coli*, sequenced, and used to transform the ΔG4 strain. Repeats in the *pglA* ORF were then mutated to avoid phase variation. A single-primer one-step mutagenesis process (Huang & Zhang, 2017) was used to generate *pglA*$_{ON*}$. Primer *plgA*_mut-fwd (Supplementary Table 2) was used to amplify the plasmid pBluescript SK (+)-*pglA*$_{ON*}$ The PCR product was digested with FastDigest DpnI at 37 °C for 2 h and then inactivated at 80 °C for 20 min. This product was transformed in XL1-Blue cells, which were then selected on kanamycin-containing LB plates. Plasmids from single colonies were sequenced and transformed in ΔG4 as described above.

**Mutagenesis of serines 112 and 114 into glycines in the C24 nanobody.** Primer C24mut_fwd (Supplementary Table 2) was used to mutate amino acids 112 and 114 on the plasmid pET26b-C24. The PCR product was digested with FastDigest DpnI at 37 °C for 2 h and then inactivated at 80 °C for 20 min. This product was transformed in XL1-Blue cells, which were then selected on kanamycin-containing LB plates. Plasmids from single colonies were sequenced and transformed in BL21 (DE3)/pLysS.

## Pili preparation

Pili were prepared as described previously (Chamot-Rooke et al. [17]). Briefly, bacteria from 10 GCB agar plates were harvested in 5 ml of 160 mM ethanolamine (Sigma-Aldrich, # 15014) at pH 10.5. Pili were sheared by vortexing for 1 min. Bacteria were centrifuged at 3200 × *g* for 45 min at 4 °C and the resulting supernatant further centrifuged at 10,000 × *g* for 30 min at ambient temperature. The supernatant was removed, pili precipitated from the suspension by the addition of 9% (vol/vol) ammonium sulfate (Sigma-Aldrich, #A4418) saturated in 160 mM ethanolamine pH 10.5 and allowed to stand for 1 h. The precipitate was pelleted by centrifugation at 1800 × *g* for 1 h at 20 °C. Pellets were washed once with cold PBS and suspended in 100 µl distilled water for mass spectrometry. For each alpaca immunization, pili were prepared from 60 GCB agar plates, dialyzed overnight against PBS with 300,000 MWCO dialysis membrane (Spectrum™, #131450 T).

## Mass spectrometry analysis

Pili preparations were resuspended in water and diluted 50 times in 20 % acetonitrile, 2 % formic acid prior to nano-infusion with a TriVersa NanoMate (Advion) into an Q Exactive HF Orbitrap mass spectrometer (ThermoFisher). Data were recorded at 15,000 resolution between 600 and 2000 m/z with 10 microscans per scan. Signals were deconvoluted using Unidec[32] with charges between 6 and 25, masses sampling every one Dalton between 15 and 20 KDa. Deconvoluted peaks in each spectrum were attributed based on the mass differences GATDH ( + 274 Da), DATDH ( + 228 Da), G3P ( + 154 Da), Hexose ( + 162 Da), HexNac ( + 203 Da).

## Nanobody libraries and phage selection

Animal procedures were performed according to the French legislation and in compliance with the European Communities Council Directives (2010/63/UE, French Law 2013-118, February 6, 2013). The Animal Experimentation Ethics Committee of Pasteur Institute (CETEA 89) approved this study (2020-27412).

One young adult male alpaca named Chunca (*Lama pacos*) was immunized at days 0, 17, and 24 with 150 µg of pili. The immunogen was mixed with Freund complete adjuvant for the first immunization and with Freund incomplete adjuvant for the following immunizations. The immune response was monitored by titration of serum samples by ELISA on pili polyclonal rabbit anti-alpaca IgGs[33].

About 250 ml of blood of the immunized animal was collected and the peripheral blood lymphocytes were isolated by centrifugation on a Ficoll (Cytiva, Velizy, France) discontinuous gradient and stored at −80 °C until further use. Total RNA and cDNA were obtained as previously described[33]. The nanobody repertoires were amplified from the cDNA by two successive PCR reactions and the nanobody fragments were cloned into the Sfi/NotI restriction sites of pHEN1 phagemid vector[34].

The selection of specific phage-nanobodies was performed by phage display (Lafaye et al. [33]). A large number of phage-nanobodies ($10^{13}$) were used to perform 3 rounds of panning. Phage-nanobodies were incubated for 1 h with the pili that has been previously coated on an immunotube (Nunc). An extensive washing procedure of the tubes was performed and specific phage-nanobodys were eluted in 100 mM triethylamine (TEA). *E. coli* TG1 at exponential growth phase were then infected with eluted phage-nanobodies. Phage-nanobodys were produced from individual colonies and binding of the phages to the pili on plate was revealed with an anti-M13 monoclonal antibody conjugated to peroxidase (Abcam). The nanobody nucleotide sequences were determined using M13-40 primer (Eurofins, Ebersberg, Germany).

## Expression of nanobodies

The coding sequences of the selected nanobodies in the vector pHEN1 were subcloned into a bacterial expression vector pET26b (Novagen) encoding an N-terminal *pelB* signal sequence plus a C-terminal His Tag

using NcoI and NotI restriction sites. Transformed *E. coli* BL21 (DE3)/pLysS cells expressed nanobodies in the periplasm after overnight induction in 2YT broth with 0.5 mM IPTG (Euromedex, #EU0008-B) at 16 °C. Bacteria were pelleted by centrifugation, resuspended in PBS buffer containing 300 mM NaCl, 1 mg/ml polymyxin B sulfate (Sigma-Aldrich, # 5291-1GM) and Complete™, EDTA-free Protease Inhibitor Cocktail (Roche, #11873580001) and incubated at 4 °C for 1 h with gentle shaking. Periplasmic extracts were obtained by centrifugation (7400 × *g*, 10 min, 4 °C). Purified nanobodies were isolated on Co++ affinity columns from periplasmic extracts, according to the manufacturer's instructions, followed by size exclusion chromatography (SEC) with a Superdex 75 column (Cytiva).

## Expression of nanobodies fused to Fc region

The coding sequences of the selected nanobodies were subcloned with the Gibson Assembly method for the expression of the dimeric nanobody-Fc fusion proteins. The nanobody genes were amplified by PCR using the primer pair pFUSE_VHH_fwd pFUSE_VHH_rev (Supplementary Table 2), and cloned with the Gibson Assembly Master Mix into pFuse-muIgG-Fc2 digested with NcoI and NotI[35]. *E. coli* XL1-Blue transformants were obtained on 2YT agar plates containing zeocin 25 µg/ml. The plasmids coding for the recombinant proteins were purified with Nucleobond Xtra Midi Plus EF (Macherey Nagel, # 740422.10), transiently transfected in Expi293™ cells (ThermoFisher Scientific) using Fectro PRO DNA transfection reagent (Polyplus), according to the manufacturer's instructions. Cells were incubated at 37 °C for 5 days and then the cultures were centrifuged. Proteins were purified from the supernatants by affinity chromatography using a HiTrap protein A HP (Cytiva), followed by SEC on a Superdex 75 column (Cytiva) equilibrated in PBS. Peaks corresponding to the dimeric nanobody-Fc proteins were concentrated and stored at −20 °C until used.

## SDS-PAGE

Polyacrylamide Gel electrophoresis (PAGE) was performed using Mini Protean TGX Stain-Free gels, 4-20 % (Bio-Rad, # 4568096) according to manufacturer's instructions. Precision Plus All Blue Protein ladder (Bio-Rad, #1610373) was used as molecular weight maker and Instant Blue (Expedeon, # ISB1L) was used to stain SDS-PAGE gel.

## Immunofluorescence of HUVECs infected cells

Primary human umbilical vein endothelial cells (HUVEC, PromoCell) were used between passages 3 and 8 and grown in Human Endothelial-SFM (Gibco) supplemented with 10% heat-inactivated FBS and 10 µg/ml of endothelial cell growth supplement (Tebubio, #BT-203) and passed every 2–3 days. $1.5.10^4$ cells were cultured in 96- well flat bottom plates (Greiner, # 655090) coated with 0.4 µg/ml rat tail type I collagen and infected the next day with *N. meningitidis* at an MOI of 100. After 30 min adhesion, unbound bacteria were washed three times with fresh cell culture medium and allowed to proliferate for 2 h. Infected cells were then fixed for 20 min in PBS containing 4% paraformaldehyde (PFA) and processed for immunofluorescence. Infected cells were incubated in blocking solution (PBS, 0.2% skin fish gelatin, PBSG) for 20 min. Samples were then incubated for 1 h with appropriate combinations of the following: 20D9 diluted in PBSG at 1 µg/ml, or nanobodies mixed with mouse anti-His tag (Biolegend, #BLE906102) diluted in PBSG at 1 µg/ml and 3.5 µg/ml respectively; except in immunofluorescence experiments involving the C24_S112-114G nanobody (Fig. 7f and g) for which concentrations of 5 µg/ml and 17.5 µg/ml were used respectively. After three washes with PBS, cells were incubated for 1 h in PBSG-containing DAPI at 0.5 µg/ml and Alexa-conjugated secondary antibodies at 10 µg/ml. After three additional washes, Mowiol was added. Images were taken with the Nikon Ti Eclipse spinning disk through a 40x oil.

## Bacterial aggregation assay

GFP-expressing *N. meningitidis* bacteria grown overnight on GCB agar plates were adjusted to an $OD_{600}$ of 0.05 and incubated for 2 h at 37 °C in RPMI medium supplemented with 10% FBS. The bacterial pre-culture was concentrated to an $OD_{600}$ of 0.3 by a centrifugation at 775 × *g* for 5 min followed by resuspension in medium. Bacterial suspensions were briefly vortexed, transferred into wells of a 96- well µ-plate with square wells (Ibidi, # 89621) and allowed to form aggregates for a period of 30 min at 37 °C in moist atmosphere containing 5% $CO_2$ either in the presence of PBS or in the presence of nanobodies at the concentrations indicated. Camelid nanobody anti-TAU was used as a negative control[36]. Four fluorescence images per well were captured using a 20x objective. The surface covered by bacterial aggregates was quantified in each field of view using a homemade macro in Fiji (Kennouche et al. [21]).

## Bacterial survival in the circulation

Experiments were performed using *N. meningitidis* strains, streaked from −80 °C freezer stock onto GCB agar plates and grown overnight in a moist atmosphere containing 5% $CO_2$ at 37 °C, and then incubated 2 more hours at 37 °C in pre-warmed RPMI-1640 medium (Gibco) supplemented with 10% FBS, at adjusted $OD_{600nm} = 0.05$, under gentle agitation. Bacteria were washed twice in 1X PBS and resuspended to $10^8$ CFU/ml in 1X PBS. SCID/Beige (CB17.Cg-*Prkdc* $^{scid}$*Lyst* $^{bg-J}$/Crl), males and females, aged between 18 and 22 weeks, were infected by retro-orbital injection of 100 µl of the bacterial inoculum ($10^7$ CFU total). Prior to infection, mice have been injected intraperitoneally with 8 mg of human transferrin (Sigma Aldrich) to allow bacterial growth in vivo. One hour after the infection, mice received a retro-orbital injection of 20 µg of nanobodies (100 µl of a preparation at 200 µg/ml), according to the group they have been randomly assigned. Blood sampling were performed at the tip of the tail at 1 h (just before treatment), 2 h and 4 h post-infection. To measure the bacteriemia, serial dilutions of the blood samples were plated on GCB agar plates and incubated overnight at 37 °C and in a moist atmosphere containing 5% $CO_2$. Bacterial counts were expressed in colony-forming units (CFU) per ml of blood and then normalized to the bacterial survival observed before the treatment. All experiments were performed in agreement with guidelines established by the French and European regulations for the care and use of laboratory animals and approved by the Institut Pasteur Committee on Animal Welfare (CETEA) under the protocol code DAP180022. For all experiments, male and female mice were used. Genders were equally balanced between groups and littermates were randomly assigned to experimental groups.

## ELISA assay

Bacteria were resuspended from fresh GCB plates in PBS at $OD_{600nm}$ of 0.2. 100 µl were dispensed within the wells of 96-well plates (Falcon, 353072#). Plates were centrifuged for 10 min at 1,600 g, and the supernatant was carefully removed. Then, plates were incubated without cover at room temperature for 30 min to allow drying. Bacteria were fixed with a solution of PBS containing 4% PFA for 15 min. Coated plates were washed 3 times with PBS and incubated in blocking solution (PBS, 0.2% gelatin, 0.1% Tween-20) for 20 min. The F10-mFc, C24-mFc and C24-mFc_S112-114G were diluted in blocking solution, added at the indicated concentrations (0.2-200 ng/ml) to the plates and incubated for 1 h. After several washes, a peroxidase-conjugated anti-mouse IgG antibody (Fc specific, Jackson ImmunoResearch, #115-036-071) diluted at 1/2,500 in blocking solution was added to the wells for 1 h. Finally, after three washes, the staining was revealed using TMB Substrate Reagent (3,3', 5,5;-tetramethylbenzidine, BD Biosciences, #555214) and stop solution following the manufacturer's instructions. The absorbance was read at 450 nm using a microtiter plate reader (Cytation 5 multimode reader (Biotek).

## Cryo-EM sample preparation and data collection

For each purified pilus sample, the concentration of the relevant T4P variant was normalized to 10 μg/ml, while the nanobodies were all at a final concentration of 50 μg/ml. 3 μl of the corresponding solution was applied to the carbon side of a lacey carbon grid and 1 μl to the back side before blotting and vitrification using a Vitrobot (ThermoFisher Scientific) at 4 °C. The glow discharge, blot time and blot force settings varied depending on the type of grid used. The grids used were Cu200-Lacey (SB-GATDH, SB-GATDH-F10, SB-GATDH C24, SA-GATDH), Au300-CFlat 1.2/1.3 (SB-DATDH-C24) and Au300-CFlat 2/2 (SB-DATDH).

For screening purposes, a 200 keV Glacios microscope was used at the Nanoimaging Core Facility of Institut Pasteur (Paris, France), equipped with either a Falcon III or a Falcon 4 detector (Thermo Fisher Scientific). Data collections at high resolution were carried out at either the Glacios or the 300 keV Titan Krios microscopes of the Nanoimaging Core Facility of Institut Pasteur, equipped with either a Gatan K3 direct electron detector and a Gatan Bioquantum LS/967 energy filter (AMETEK) or a Falcon 4i detector and a SelectrisX energy filter (Thermo Fisher Scientific). Images were acquired using the EPU software (Thermo Fisher Scientific).

All the data acquisition parameters, the number of movies acquired as well as the microscope and camera setup utilized for each of the samples, can be consulted in Supplementary Table 1.

## Cryo-EM image processing and 3D reconstruction

The SB-GATDH and SB-GATDH-F10 datasets were motion corrected, dose weighted and CTF estimated using the MotionCor2[37] and GCTF[38] wrappers within the Relion 3 environment[39]. Particle processing was also carried out in the cryoSPARC2 and cryoSPARC3 environments[40]. The remainder of the datasets were preprocessed with the Patch Motion Correction and Patch CTF functionalities, and then fully processed inside the cryoSPARC3 and cryoSPARC4 softwares.

In all the datasets, an initial manual particle picking step was followed by iterative filament tracer jobs within cryoSPARC using subsequently better 2D class averages as templates. The hysteresis threshold (value for true edge detection) and minimum filament length parameters were custom to account for filament overlapping, bending or intersecting. In addition, a very restrictive filament curvature and sinuosity threshold (filament length to total contour ratio) was enforced on the picked particles prior to their extraction. Final particles were extracted with different box sizes, always at least 4 times the predicted filament diameter. The number of particles extracted and the number of pixels per box for each dataset can be consulted in Supplementary Table 1. Particles were cleaned by several rounds of 40 or 50-class 2D classification steps with increasing T values and box corner masking.

In the case of the SB-GATDH pilus, the average power spectra of the dataset were determined in cryoSPARC using the selected particles. Then, HELIXPORER-1 (https://rico.ibs.fr/helixplorer/) was used to determine initial twist, rise, and number of units per turn parameters (Twist = 102.528 deg, rise = 10.26 Å, 3.5 units/turn). A first helical refinement job was run following this estimation, with an assymetric initial model and wide searches of twist, rise and helical symmetry order. In all the other datasets, the values from the final SB-GATDH refinement were used as initial exploratory values.

In subsequent refinement cycles, the initial volume was the map output from the previous refinement, low-pass filtered to 15 Å and all the search parameters were more restrictive. In the final helical refinement step, CTF-refined particles were used and non-uniform refinement was enabled, along with FSC-noise substitution and the use of a tight dynamic mask. In the SB-DATDH and SB-DATDH-C24 datasets, the glycosylation density required an increase in local resolution. Therefore, in those cases a symmetry expansion step was performed after helical refinement followed by a local refinement step with a Z-clipped mask that was 50% of the length of the map and covered the full length of a PilE monomer.

The final sharpening of all EM maps was performed by unbiased tight-target neural network inference using the DeepEMhancer program[38].

The global resolution of the final map of all the reported structures was estimated from half-maps according to the gold standard Fourier shell correlation (GS-FSC) with a cutoff of 0.143[41]. The FSC cutoff of 0.5 was also reported by interpolation using the curve data.

EM density maps obtained were visualized with UCSF Chimera[42] and Coot 0.9.4[43].

## Model building and refinement

Models for fitting and refinement of the nanobodies were originally generated by de-novo prediction using AlphaFold2[44], starting with the FASTA sequences of the relevant proteins. The pLDDT score was used as a reference since it is recommended for single protein models, and finally the top-ranked model was selected for docking from each prediction. For the SB-GATDH protomer modeling, PDB entry 5KUA[18] was used as a starting point, with following models using SB-GATDH as initial reference.

To start, the models were rigid-body fit close to their densities within ChimeraX[45] and then were docked with jiggle-fit by Fourier filter within Coot. They were then roughly built by hand using real-space refinement with the sharpened map as guide. Further refinement was first performed with real-space refinement within Phenix[46].

Ligands GATDH (GAT), DATDH (DAT) and Glycerol-3-Phosphate (G3P) were then generated from their SMILES sequence with the JLigand[47] program within the CCP4[48] software suite, along with restraints files when linked to Serine residues. All ligands were roughly fit on Coot and the OG of Ser69 was deleted in all cases, and subsequently the entire complexes were refined for 20 macro cycles on Phenix with all the .cif restraint files and custom bond definition .phil files for Ser63-GAT/DAT and Ser69-G3P links. Validation rounds were perfomed in Molprobity[49] within the Phenix package. All the model refinement statistics for the 6 datasets are presented in Supplementary Table 1.

## Molecular dynamics

**System preparation.** GATDH, DATDH, and G3P were parametrized stating from existing structures in the charm forcefield and applying the existing patches for group modification and covalent binding to the appropriate serine residues (e.g. SGPB patch for B-glycosylation of SER). The system was built by inserting a Type IV Pilus 20-mer model into a 10 nm x 10 nm 100% POPE (1-palmitoyl-2-oleoyl-sn-glycero-3-phosphoethanolamine) membrane, a model used before for similar systems[50,51], and close to the *N. meningitidis* IM composition[52], using the CHARMM-GUI Membrane Builder server[53]. For Chain A, at the bottom of the pilus model, the N-terminus (Phe1-Tyr27) was modeled into a completely folded alpha-helix -as previously observed in other *N. meningitidis* pilin structures, e.g. 2PIL 2HI2[25,54] based on the AF-A0A1I9GEU1-F1 AlphaFold model obtained from UniProt entry A0A1I9GEU1, so as to allow a better insertion of the other subunits presenting "free" N-termini, not embedded into the pilus core by other subunits' globular domains, into the membrane as well. The protein was initially placed into the membrane with the refolded alpha-helix of Chain A inserted up to P22 into the membrane, and the flexible linker portion of chain B partially inserted into the bilayer. The TIP3P water model was used for hydration, choosing to leave a 2 nm water layer above/below the model, creating a rectangular box of ~10 ×10 x 31.5 nm. Sodium Chloride was added to a concentration of 0.15 M and temperature was set to 310 K, to mimic physiological conditions. The systems consist of ~310000 particles.

**Molecular dynamics simulations.** MD simulations were performed with Gromacs 2022.5[55] interlaced with PLUMED 2.9.0[56] using the CHARMM36m forcefield[57]. Energy minimization was carried out until convergence using a steepest descents algorithm for 5000 steps. The system was then equilibrated in 5 cycles, gradually reducing constraints, and increasing time-step, as suggested by the CHARMM-GUI server, using the leap-frog algorithm, Verlet cut-off scheme for VdW interactions gradually switched off at 10–12 Å, and PME for electrostatics, with a 1.2 nm cutoff, and the Berendsen thermostat at 310 K discriminating between 3 groups, protein, solvent and membrane. Hydrogens were constrained using LINCS, a semiisotropic Berendsen barostat was used starting from the third equilibration step. Production was then performed, for 1000 ns, using the same parameters except that a Nose-Hoover thermostat was used as well as a semiisotropic Parrinello-Rahman barostat. Coordinates were written every 10 ps. Simulations were run by triplicate.

**MD trajectory analysis.** Analysis was performed using the different GROMACS utilities (e.g. gmx rms, gmx rmsf, and gmx distance). Only the central 10 subunits of the pilus were considered for analysis, excluding 10 subunits from the top and the bottom. RMSD analysis of backbone atoms suggested subsequent analysis to be conducted after the system reaches stability, i.e. discarding the first ~100 ns. RMSF analysis of the whole protein was computed for Backbone atoms (except when analyzing the sugar molecules), with respect to the average conformation. UCSF ChimeraX[42] developed by the Resource for Biocomputing, Visualization, and Informatics at the University of California, San Francisco, with support from National Institutes of Health R01-GM129325 and the Office of Cyber Infrastructure and Computational Biology, National Institute of Allergy and Infectious Diseases was used for image, and VMD[58] was used for visualisation of trajectories and videos. Graphs were generated with Xmgrace (Turner, P. J. "XMGRACE, Version 5.1. 19." Center for Coastal and Land-Margin Research, Oregon Graduate Institute of Science and Technology, Beaverton, OR 2 (2005).

### Reporting summary
Further information on research design is available in the Nature Portfolio Reporting Summary linked to this article.

## Data availability
Source data are provided with this paper. The cryo-electron microscopy data generated in this study have been deposited in the Electron Microscopy Databank under accession codes: EMD-17375 (SB-GATDH structure), EMD-17384 (SB-DATDH structure), EMD-17718 (SB-GATDH-F10 structure), EMD-17683 (SB-GATDH-C24 structure), EMD-17695 (SB-DATDH-C24 structure) and EMD-17386 (SA-GATDH structure). The structural data generated in this study have been deposited in the Protein Data Bank under accession codes 8P2V (SB-GATDH structure), 8P36 (SB-DATDH structure), 8PJP (SB-GATDH-F10 structure), 8PIJ (SB-GATDH-C24 structure), 8PIZ (SB-DATDH-C24 structure) and 8P3B (SA-GATDH structure). All mass spec data (raw and processed) are available via ProteomeXchange with accession code PXD046119 Source data are provided with this paper.

## Code availability
The input files, parameters and final snapshots for the Molecular Dynamics carried out in this study have been deposited in the Zenodo repository under accession code 10637491 (SB-GATDH, SB-DATDH, SA-GATDH MDs).

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

## Acknowledgements

We would like to warmly thank Sandrine Moutel (Institut Curie) for kindly providing the pFuse vector; and Pablo Guardado, Pedro Alzari and Ahmed Haouz for critical reading of the manuscript. This work was supported by the Integrative Biology of Emerging Infectious Diseases (IBEID) laboratory of excellence (ANR-10-LABX-62), Prix Fondation NRJ-Institut de France 2021 (G.D.), la Fondation pour la Recherche Médicale (FRM, EQU202203014654 to G.D.), D.F.M. was supported by a grant from the Agence Nationale de la Recherche (ANR 18 CE11 0022 T4PNA-NOACTION to G.D.). A.Z. was supported by a grant from DARRI and Institut Carnot (CONSO INNOV 31-19).

## Author contributions

D.F.M., Y.K and G.D. conceived the project and designed the experiments; D.F.M., Y.K and S.G. performed experiments; D.F.M., Y.K and A.M. solved the cryoEM structures; A.Z. performed molecular dynamics approaches; P.G. performed in vivo experiments; M.R. and J.C.R. performed and analyzed MS experiments; G.A., P.L. and S.G. isolated and produced nanobodies; M.V. contributed to image acquisition. All authors contributed to the interpretation of the data. D.F.M. and G.D. wrote the manuscript and all authors read and approved the final manuscript.

## Competing interests

The authors declare no competing interest.

## Additional information

**Peer review information** : *Nature Communications* thanks Xiongwu Wu and the other, anonymous, reviewer(s) for their contribution to the peer review of this work. A peer review file is available.

