## [Peer Review File · Nature Communications]

Cryo-EM structures of type IV pili complexed with nanobodies reveal immune escape mechanismsREVIEWER COMMENTS

Reviewer #1 (Remarks to the Author):

This is an interesting combination of cryoEM, MD and functional studies focused on the mechanisms of *Neisseria meningitidis* to introduce variations in the surface of its pilus, which helps to explain why thus far the search for an antibody against the pilus had little success. The authors present first three cryo-EM structures of Type IV pilus (T4P) class I corresponding to three subvariants. In addition, they generated two nanobodies against T4P and obtained three additional high-resolution cryo-EM structures of two of the subvariants with a nanobody bound. The cryo-EM structures appear to have good resolution which resolved sequence variations and posttranslational modifications, supporting the conclusions. Novel ideas are that the combination of sequence variations and post translational modifications alters the pilus surface, which affects antibody recognition, and that a combination of camelid nanobodies could be a successful strategy to combat immune evasion. Overall, the reviewer thinks that the manuscript is worthy of publication in *Nature Communications* but there remain concerns on the quality of the figures as well as the precision of the writing as indicated below that should be addressed.

1. The manuscript is not written for the general public as the authors rely heavily on the jargon in the field. Please edit to attract the interest of the more general reader.
2. The introduction is not well focused and lacks clarity. It should state that only class I will be examined, which variants, and why.
3. Methods: the number of experimental replica are not indicated in most sections, this information needs to be included.
4. Results: The six reconstructions reach good resolution and support the results, but several pieces of information are missing:
 - a) pilus geometry: is it 3-start helix or 4-start helix, or another geometry?
 - b) # of filaments analyzed and # of filaments discarded for each dataset.
 - c) Please specify whether the 2D classification produce different classes, or was 3D used to select the best classes?

5. Figures are a bit preliminary, they need improvement as described below.

Fig 2 b-d: needs a better figure legend and definition of the scale bar.

Fig 4a: please expand the boundary of the pili, the current cropping does not allow seeing the protruding antibodies.

Fig 5: please add a pilus and highlight which sections are shown in a-e

Fig 6a,b: please separate the two sets of structures, it is very confusing to discern the two forms

Fig 6c: please change the colors or enlarge the image: the blue, red and violet are difficult to distinguish

Fig. S2a: The panel with initial helical symmetry estimation has low contrast; diffraction spots are barely seen. Please improve the contrast

Fig S2c: define better what is meant by a "middle section of the pilus" as it could lead to confusion

Fig S5: graph is missing the x-axis label. On the results, please comment on the intermediate species around the tall peaks

Fig S6 is missing

6. The presentation of the Molecular Dynamics data should not only rely on movies; please make figures relaying the main findings.

7. Many sentences in the discussion are too vague which reduces the interest and does not convey much information. Examples of such sentences are "could impact antibody binding", "significantly impacts antibody recognition "(but how?), has "strong implication", "substantial influence", etc. Please substitute with informative sentences.

OTHER SPECIFIC COMMENTS:

Line 230: "In the case of F10, CDR1 does not seem to contribute significantly to the F10 binding, it remains on the side of the nanobody." Please define CDR and as it is part of the nanobody, it is not surprising that remains on its side; probably the authors mean something else.

Line 319: "Interestingly strand b1 of the beta sheet strand parallel to the alpha-beta loop is also submitted to several amino acid changes the importance of which was previously not recognized." Improve the sentence and add the residue range for clarity.

Line 340-342: “The structure of the F10-pilus complex shows the high specificity of the VHH for the SB hypervariable loop, but in exchange this VHH avoids variability in the region close to G3P.” Please improve sentence.

Lines 662, 664: please describe the “hysteresis threshold” and “sinuosity threshold”.

Page 45 does not seem to belong to the manuscript.

Reviewer #2 (Remarks to the Author):

In the manuscript “Cryo-EM structures of type IV pili complexed with nanobodies reveal immune escape mechanisms” the authors provide a detailed structure-based analysis of *Neisseria meningitidis* immune escape mechanisms by obtaining cryoEM structures of type IV pilus variants in the absence and presence of bound nanobodies. The cryoEM reconstructions are high quality, at resolutions suitable to identify posttranslational modifications and amino acid differences that contribute to immune escape. They show how antibodies bind to variable surfaces like the PTMs and hypervariable loop of the major pilin, and how small changes in these can disrupt binding. They further show that these anti-pilin nanobodies, when fused with mouse Fcs, can prevent *N. meningitidis* growth in the mouse model, demonstrating how an understanding of these molecular processes can be of potential therapeutic value. Given the hypervariability in the major pilin the authors might comment in the Discussion on how the type IV pilus is such an effective adhesion. Most of my additional comments are editorial in nature.

Line 61-63 – It should be noted that not all type IV pili exhibit surface variation.

Line 66072 – Say something about the prevalence of Class I vs II strains (as in Line 355).

Line 99 – I recommend using “nanobodies” instead of the cryptic “VHH” acronym.

Line 110 – “isolated segments extracted and curated” Please explain “curated” and indicate that the filaments that show well resolved structural features are class averages, not individual pilus filaments. Similarly, in the legend for the figure in question, (line 770)– “2D classes” should be “2D class averages”. In this same line remove “high resolution” as pilin monomers and secondary structure are not high (i.e. atomic) resolution features.

Line 131 – OG, OE1 – use Greek letters for G and E here and elsewhere.

Line 134 – “cycle” should be “ring”

Line 142-153 – Clarify that the only differences between SA and SB pili are in sequence, not PTMs.

Line 167 – Indicate from which strain the pili were purified - 8013 SB? Cite Fig. S1c instead of S1b.

Line 174 – Change “the same procedure was conducted” to “binding was also tested” (“procedure” sounds like the phage selection procedure).

Line 197-197 – Change “the effect of nanobodies on a possible neutralization of” to “the ability of nanobodies to neutralize”.

Line 199 – Report nanobody concentration using the same units here and in the figure.

Line 202 – For clarity include “phagocytic”, i.e. “phagocytic Kupffer cells” (and correct the spelling of Kupffer here and in Line 355).

Line 270 – Delete “which is not a variable area”.

Line 277 – Change “cycle” to “ring”.

Line 312 – “aminoacidic” should be “amino acid”

Line 320 – Change “is also submitted to” to “undergoes”.

Line 336 – Delete “for instance”.

Line 353 – “implication in” should be “effect on”

Line 361 – delete “Nevertheless,”

Line 376 – “pilus head surface” – should this be “pilin surface” or “exposed pilin surface”?

Line 383 – Not clear what “G/ATDH” or this sentence means – reword.

Line 404 – delete “specific”

Line 474 – “during” should be “for”

Line 484 – “Preparation of pili” should be “Pilus preparations” or just “Pili”.

Line 489 – “pics” should be “peaks”

Line 498 – Is the alpaca’s name relevant? (and Alpaca need not be capitalized).

Line 517 – Correct spelling of peroxidase.

Line 585 – “Camelidae” should be “Camelid”.

Line 749/50 - Check period placements.

Line 775 – Variable amino acids are shown in green – is this regardless of whether or not the changes are conserved? Recommend coloring only non-conserved changes.

Line 823 – Correct “phosphoglycerol”.

Line 844 – “indicate”

Fig. S4 – Some of the text is small and faint so hard to read.

Supplementary tables are not labeled. Something is wrong with the formatting.

Reviewer #3 (Remarks to the Author):

This work produced the cryo-electron microscopy structures of pili of different sequence types with sufficiently high resolution to visualize posttranslational modifications and their complexes with nanobodies. Molecular dynamics simulations contributed to the structure modeling and refinement.

While little simulation data was presented to validate the findings, the supplementary videos provide some details. The simulation protocol seems alright. The video S1 shows the membrane with a 16-mer pilus inserted keeps moving up, or the 16-mer keeps inserting deeper, until the final quarter of the simulation when G3P touches the membrane. This does not invalidate the simulation, but does show that the initial insertion of the pilus is not right.

Point by point response for Nature Communications manuscript NCOMMS-23-47033-T

REVIEWER COMMENTS

Reviewer #1 (Remarks to the Author):

This is an interesting combination of cryoEM, MD and functional studies focused on the mechanisms of *Neisseria meningitidis* to introduce variations in the surface of its pilus, which helps to explain why thus far the search for an antibody against the pilus had little success. The authors present first three cryo-EM structures of Type IV pilus (T4P) class I corresponding to three subvariants. In addition, they generated two nanobodies against T4P and obtained three additional high-resolution cryo-EM structures of two of the subvariants with a nanobody bound. The cryo-EM structures appear to have good resolution which resolved sequence variations and posttranslational modifications, supporting the conclusions. Novel ideas are that the combination of sequence variations and post translational modifications alters the pilus surface, which affects antibody recognition, and that a combination of camelid nanobodies could be a successful strategy to combat immune evasion. Overall, the reviewer thinks that the manuscript is worthy of publication in Nature Communications but there remain concerns on the quality of the figures as well as the precision of the writing as indicated below that should be addressed.

1. The manuscript is not written for the general public as the authors rely heavily on the jargon in the field. Please edit to attract the interest of the more general reader.

Efforts have been made to reduce topic-specific language and make the text accessible to scientist from other areas. In particular, the beginning of the introduction has been rewritten to fit a broader public.

2. The introduction is not well focused and lacks clarity. It should state that only class I will be examined, which variants, and why.

Introduction has been modified to be better define objectives and to explain the choices of variant.

3. Methods: the number of experimental replica are not indicated in most sections, this information needs to be included.

This is now indicated in the figure legends.

4. Results: The six reconstructions reach good resolution and support the results, but several pieces of information are missing:

a) pilus geometry: is it 3-start helix or 4-start helix, or another geometry?

The type-IV pili described in the paper follow three types of helical geometries: A 1-start right-handed (+1) helix, 3-start left-handed (-3) and a 4-start right-handed geometries. These geometries were also originally described by Kolappan and colleagues (Nat Com 2016). A reference to the helical geometries and this publication is now included.

b) # of filaments analyzed and # of filaments discarded for each dataset.

The particles used and discarded can be consulted in Supplementary Table 1, along with collection details and refinement statistics. The table has been modified for better clarity.

c) Please specify whether the 2D classification produce different classes, or was 3D used to select the best classes?

2D classification was extensively used to clean the particles and yielded well-centered, vertically aligned homogeneous filaments or diverse orientations. In all cases, heterogeneous refinement or 3D classification attempts did not yield alternative conformations or large quantities of low-quality particles, therefore virtually all particles selected during the 2D classification step were used for helical refinement. The text has been modified accordingly.

5. Figures are a bit preliminary, they need improvement as described below.

Fig 2 b-d: needs a better figure legend and definition of the scale bar.

Modified

Fig 4a: please expand the boundary of the pili, the current cropping does not allow seeing the protruding antibodies.

Modified

Fig 5: please add a pilus and highlight which sections are shown in a-e

A pilin monomer has been added with an indication the regions shown in a-e

Fig 6a,b: please separate the two sets of structures, it is very confusing to discern the two forms

Structures have been split into different panels for more clarity

Fig 6c: please change the colors or enlarge the image: the blue, red and violet are difficult to distinguish

The panels showing the cryoEM maps have been doubled in size. Fig 6 has been split into 2 figures and it now appears as Fig 7a.

Fig. S2a: The panel with initial helical symmetry estimation has low contrast; diffraction spots are barely seen. Please improve the contrast

The resolution and contrast have been improved and image made larger.

Fig S2c: define better what is meant by a “middle section of the pilus” as it could lead to confusion

The sentence has been reworded to clarify

Fig S5: graph is missing the x-axis label. On the results, please comment on the intermediate species around the tall peaks

The axis label is now included and intermediate peaks commented in the figure legends section.

Fig S6 is missing

This is corrected. Actually, there was no Fig. S6 intended in the version sent out for review. A Fig. S6 legend was left by mistake from initial versions.

6. The presentation of the Molecular Dynamics data should not only rely on movies; please make figures relaying the main findings.

Quantitative analysis of the Molecular Dynamics have been included in the manuscript. A supplementary figure (Fig. S3) was added to show two quantitative measurements:

- **RMSD of the overall structure during 1 μ s molecular dynamics showing the stability of the structures and validating the duration of the MD (Fig S3a)**
- **RMSF of the structures quantifying the large movements of the SB sequence type hypervariable region (Fig S3b)**

Furthermore, the frequency of the hydrogen bond formed between the GATDH and E56 has been determined and indicated in the main text.

7. Many sentences in the discussion are too vague which reduces the interest and does not convey much information. Examples of such sentences are “could impact antibody binding”, “significantly impacts antibody recognition “(but how?), has “strong implication”, “substantial influence”, etc. Please substitute with informative sentences. **Several modifications have been made in the discussion to address this point.**

OTHER SPECIFIC COMMENTS:

Line 230: “In the case of F10, CDR1 does not seem to contribute significantly to the F10 binding, it remains on the side of the nanobody.” Please define CDR and as it is part of the nanobody, it is not surprising that remains on its side; probably the authors mean something else.

The acronym CDR was spelled out and sentence clarified.

Line 319: “Interestingly strand b1 of the beta sheet strand parallel to the alpha-beta loop is also submitted to several amino acid changes the importance of which was previously not recognized.” Improve the sentence and add the residue range for clarity.

This point was clarified

Line 340-342: “The structure of the F10-pilus complex shows the high specificity of the VHH for the SB hypervariable loop, but in exchange this VHH avoids variability in the region close to G3P.” Please improve sentence.

The sentence was rephrased to clarify

Lines 662, 664: please describe the “hysteresis threshold” and “sinuosity threshold”.

A definition has been included in the text

Page 45 does not seem to belong to the manuscript.

I am not sure which page this corresponds to, there does not seem to be any extra page in the PDF that was included on our side, perhaps this results from the submission system?

Reviewer #2 (Remarks to the Author):

In the manuscript “Cryo-EM structures of type IV pili complexed with nanobodies reveal immune escape mechanisms” the authors provide a detailed structure-based analysis of *Neisseria meningitidis* immune escape mechanisms by obtaining cryoEM structures of type IV pilus variants in the absence and presence of bound nanobodies. The cryoEM reconstructions are high quality, at resolutions suitable to identify posttranslational modifications and amino acid differences that contribute to immune escape. They show how antibodies bind to variable surfaces like the PTMs and hypervariable loop of the major pilin, and how small changes in these can disrupt binding. They further show that these anti-pilin nanobodies, when fused with mouse Fcs, can prevent *N. meningitidis* growth in the mouse model, demonstrating how an understanding of these molecular processes can be of potential therapeutic value. Given the hypervariability in the major pilin the authors might comment in the Discussion on how the type IV pilus is such an effective adhesion. Most of my additional comments are editorial in nature.

Line 61-63 – It should be noted that not all type IV pili exhibit surface variation. **In pathogenic *Neisseria* species all strains express pili with potential surface variation, due to amino acid changes and/or PTMs such as sugars. Type IV pili from other bacterial species can indeed exhibit no or moderate surface variation. The text was modified to clarify that statements referred to *Neisseria* spp.**

Line 66072 – Say something about the prevalence of Class I vs II strains (as in Line 355).

A sentence has been added.

Line 99 – I recommend using “nanobodies” instead of the cryptic “VHH” acronym. **The term “VHH” was replaced by “nanobody” throughout the manuscript**

Line 110 – “isolated segments extracted and curated” Please explain “curated” and indicate that the filaments that show well resolved structural features are class averages, not individual pilus filaments. Similarly, in the legend for the figure in question, (line 770)– “2D classes” should be “2D class averages”. In this same line remove “high resolution” as pilin monomers and secondary structure are not high (i.e. atomic) resolution features.

The sentence has been reworded and the suggested corrections in the legend made

Line 131 – OG, OE1 – use Greek letters for G and E here and elsewhere.

Greek letter have been introduced

Line 134 – “cycle” should be “ring”

Replaced

Line 142-153 – Clarify that the only differences between SA and SB pili are in sequence, not PTMs.

A sequence was added to clarify this important point

Line 167 – Indicate from which strain the pili were purified - 8013 SB? Cite Fig. S1c

instead of S1b.

This is indicated

Line 174 – Change “the same procedure was conducted” to “binding was also tested” (“procedure” sounds like the phage selection procedure).

Done

Line 197-197 – Change “the effect of nanobodies on a possible neutralization of” to “the ability of nanobodies to neutralize”.

Change made

Line 199 – Report nanobody concentration using the same units here and in the figure.

This is now the case

Line 202 – For clarity include “phagocytic”, i.e. “phagocytic Kupffer cells” (and correct the spelling of Kupffer here and in Line 355).

Modifications included

Line 270 – Delete “which is not a variable area”.

Deleted

Line 277 – Change “cycle” to “ring”.

Changed

Line 312 – “aminoacidic” should be “amino acid”

Corrected

Line 320 – Dchange “is also submitted to” to “undergoes”.

Change made

Line 336 – Delete “for instance”.

Deleted

Line 353 – “implication in” should be “effect on”

Change made

Line 361 – delete “Nevertheless,”

Deleted

Line 376 – “pilus head surface” – should this be “pilin surface” or “exposed pilin surface”?

This sentence was rephrased accordingly.

Line 383 – Not clear what “G/ATDH” or this sentence means – reword.

We meant GATDH or DATDH but GATDH is sufficient here and the text was corrected

Line 404 – delete “specific”

Deleted

Line 474 – “during” should be “for”

Modified

Line 484 – “Preparation of pili” should be “Pilus preparations” or just “Pili”.

Changed

Line 489 – “pics” should be “peaks”

Corrected

Line 498 – Is the alpaca’s name relevant? (and Alpaca need not be capitalized).

This is probably not very important but the identity of the alpaca could be relevant considering that, over the years, these animals receive successively several injections with different antigens and they have their own immunological history.

Line 517 – Correct spelling of peroxidase.

Corrected

Line 585 – “Camelidae” should be “Camelid”.

Changed

Line 749/50 - Check period placements.

Corrected

Line 775 – Variable amino acids are shown in green – is this regardless of whether or not the changes are conserved? Recommend coloring only non-conserved changes.

Sentence has been changed to clarify

Line 823 – Correct “phosphoglycerol”.

Corrected

Line 844 – “indicate”

Corrected

Fig. S4 – Some of the text is small and faint so hard to read.

Text has been made larger

Supplementary tables are not labeled. Something is wrong with the formatting.

The table and its format have been modified accordingly

Reviewer #3 (Remarks to the Author):

This work produced the cryo-electron microscopy structures of pili of different sequence types with sufficiently high resolution to visualize posttranslational modifications and their complexes with nanobodies. Molecular dynamics simulations

contributed to the structure modeling and refinement.

While little simulation data was presented to validate the findings, the supplementary videos provide some details. The simulation protocol seems alright. The video S1 shows the membrane with a 16-mer pilus inserted keeps moving up, or the 16-mer keeps inserting deeper, until the final quarter of the simulation when G3P touches the membrane. This does not invalidate the simulation, but does show that the initial insertion of the pilus is not right.

The pilus is actually not sinking into the lipid bilayer, rather, the pilus occasionally tilts to the side giving the impression that it is sinking depending on the angle of observation. A different angle was chosen to avoid this impression.

REVIEWERS' COMMENTS

Reviewer #1 (Remarks to the Author):

The authors addressed my concerns satisfactorily.

One exception is the revised Supplementary Table S1. It was not possible to evaluate it as there was no updated Supplementary PDF file file besides the files for individual supplemental figures.

Reviewer #2 (Remarks to the Author):

Concerns have been addressed to my satisfaction. In my opinion the manuscript is suitable for publication.

Reviewer #3 (Remarks to the Author):

The authors' response addressed my question about simulation setup. No more questions about this manuscript.